# Spectroscopic and Colorimetric Studies for Anions with a New Urea-Based Molecular Cleft

**Sanchita Kundu, Tochukwu Kevin Egboluche, Zehra Yousuf and Md. Alamgir Hossain ***

Department of Chemistry, Physics and Atmospheric Sciences, Jackson State University, Jackson, MS 29217, USA; sanchita.v23@gmail.com (S.K.); tochukwukevin1986@gmail.com (T.K.E.); zayousuf9@gmail.com (Z.Y.)
* Correspondence: alamgir.hossain@jsums.edu

**Abstract:** A new simple urea-based dipodal molecular cleft (**L**) has been synthesized and studied for its binding affinity for a variety of anions by $^1$H-NMR, UV-Vis and colorimetric techniques in DMSO-$d_6$ and DMSO, respectively. The results from titration studies suggest that the receptor forms a 1:2 complex with each of the anions used via hydrogen bonding interactions and exhibits strong selectivity for fluoride among halides, showing the binding affinity in the order of fluoride > chloride > bromide > iodide; meanwhile, it displays moderate selectivity for acetate among oxoanions, showing the binding affinity in the order of acetate > dihydrogen phosphate > bicarbonate > hydrogen sulfate > nitrate. Colorimetric studies of **L** for anions in DMSO reveal that the receptor is capable of detecting fluoride, acetate, bicarbonate and dihydrogen phosphate, displaying a visible color change in the presence of the respective anions.

**Keywords:** chemosensor; anion recognition; neutral receptor; urea; molecular cleft

## 1. Introduction

Anions are ubiquitous in nature and play an important role in biology, medicine, catalysis, agriculture and in many other industries [1–4]. Halides and oxoanions are essential in many biological processes that are key components in life to properly function for growth, reproduction and genetic signaling [5]. Many anions are associated with developing personal care products, pharmaceuticals, fertilizers and constructions materials, fire extinguishers, etc. [6]. However, when the concentration of anions becomes higher than the normal permissible range, they may adversely affect both biological systems as well as the environment. For example, an elevated amount of fluoride in drinking water causes dental and skeletal fluorosis [7] and is responsible for a bone cancer known as osteosarcoma [8]. The presence of an excess phosphate in the environment leads to eutrophication [9,10] of natural water bodies deteriorating normal aquatic life. A high level of phosphate in the human body increases the risk of hyperparathyroidism [5], soft tissue calcification, cardiovascular complications [11] and crystal deposition disease [12]. Therefore, the detection and binding of the key inorganic anions in vitro or in vivo is much anticipated. The remediation of an excess anion is very important for maintaining a healthy balance between the biological and environmental systems, leading to an upsurge in attention in synthesizing efficient artificial anion sensors [13,14] in supramolecular chemistry in recent decades [15–20]. However, the strategic design and synthesis of functional anion receptors showing measurable physico-chemical change upon the reaction with anions to produce signal still remain a major challenge. Several factors, including a larger size of anions compared to their isoelectronic cations, diverse geometries (halides: spherical; carbonate, nitrate, acetate: trigonal planar; phosphate, sulfate: tetrahedral) and pH sensitivity [21,22], restrict their interactions with the target receptors. Furthermore, the lesser charge-to-size ratio of anions as compared to their cationic counterparts inhibits their ability for electrostatic interactions with synthetic receptors. The high solvation energies are an added impediment in efficient sensing, making it difficult for the receptors to overcome

the solvation barrier created by the solvent used. Polar solvents, such as water, also tend to compete with anionic analytes for binding with receptors.

In recent years, a substantial amount of work on urea-based and thiourea-based neutral anion receptors has been reported in the literature [23–28]. Molecules with certain functional groups, such as urea, thiourea, amides and other NH-bearing derivatives, are often introduced in the structural designing of chemosensors [29–37] to facilitate efficient binding with anions due to their directional hydrogen-bonding ability with anions [38–40]. The increased polarity of an NH unit owing to the presence of electron-withdrawing groups in the structure further enhances an anion binding event offering strong H-bonding interactions [41]. An ideal cleft-based chemosensor should possess active binding sites linked with a suitable spacer to create a naked cavity functionalized with signaling units (Scheme 1), thus ensuring to stabilize an encapsulated complex by essential binding forces. Previously, we reported *m*-xylyl-based molecular clefts possessing confined cavities that showed excellent anion binding capabilities in solution [8]. In an attempt to explore selective colorimetric probes [22,42–44], we synthesized a simple *para*-xylyl-based dipodal molecular urea **L** containing a flexible cavity that is appended with two aryl nitro-groups as signaling units (Scheme 2). Herein is reported the synthesis of a molecular cleft **L** and its binding affinity towards different halides and oxoanions in DMSO, displaying a visible color change in the presence of certain anions.

**Scheme 1.** Schematic of molecular cleft and its anion complex.

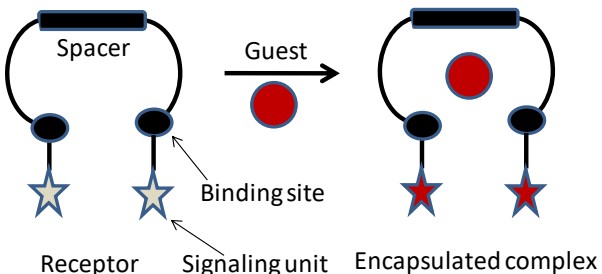

**Scheme 2.** Schematic diagram of the synthesis of **L**.

## 2. Materials and Methods

### 2.1. Materials

All reagents and solvents were purchased from commercial suppliers and were used without further purification. Nuclear magnetic resonance (NMR) spectra were recorded using a DMSO-$d_6$ solvent at 25 °C on a Varian Unity INOVA 500 FT-NMR instrument. The chemical shifts were measured in parts per million (ppm) using tetramethylsilane as a reference. All NMR data were processed and analyzed with MestReNova software. The UV-Vis screening and titration experiments were performed in DMSO at room temperature using a UV-2600 spectrophotometer (SHIMADZU). Elemental analysis was carried out using an ECS 4010 Analytical Platform (Costech Instrument, Valencia, CA, USA) at Jackson State University.

## 2.2. Synthesis

The receptor **L** was synthesized from a reaction of *p*-xylylenediamine (0.27 g, 2.0 mmol) and 3-nitrophenyl isocyanate (0.66 g, 4.0 mmol) in a 1:2 stoichiometric ratio in dry acetonitrile under constant stirring at room temperature. After 24 h of continuous stirring, the light-yellow precipitate (crude **L**) was collected by filtration. The yellowish residue was then purified by washing thoroughly with acetonitrile and THF, followed by drying under vacuum to obtain the target receptor **L** (0.65 g, 70% yield). $^1$HNMR (500 MHz, DMSO-$d_6$) δ (ppm): 6.80 (s, 2H, NH1), 9.11 (s, 2H, NH2), 8.50 (s, 2H, H3), 7.64–7.62 (d, 2H, J∼10.0 Hz, Ar-H4), 7.50–7.48–7.46 (t, 2H, J∼10.0 Hz, Ar-H5), 7.73–7.72 (d, 2H, J∼5.0 Hz, Ar–H6), 4.28–4.27 (d, 4H, J∼5.0 Hz, H7), 7.25 (s, 4H, Ar-H8). $^{13}$CNMR (125 MHz, DMSO-$d_6$, TSP): δ 127.60 (Ar–$C_a$), 142.22 (Ar-$C_b$), 42.97 (Alph-$C_c$), 155.36 (Alph–$C_d$), 139.03 (Ar–$C_e$), 111.99 (Ar–$C_f$), 148.52 (Ar-$C_g$), 115.97 (Ar-$C_h$), 130.35 (Ar-$C_i$), 124.16 (Ar-$C_j$). Anal. Calcd. for $C_{22}H_{20}N_6O_6$: C, 56.89; H, 4.34; N, 18.10; O, 20.67. Found: C, 56.82; H, 4.30; N, 18.07; O, 20.65.

## 2.3. UV-Vis Titration Studies

The receptor (**L**) was titrated spectroscopically by UV/Vis spectrometry against various halides and oxoanions ($F^-$, $Cl^-$, $Br^-$, $I^-$, $HSO_4^-$, $H_2PO_4^-$, $HCO_3^-$, $AcO^-$ and $NO_3^-$) using tetrabutyl ammonium salts in DMSO. In this experiment, a stock solution of **L** ($5 \times 10^{-3}$ M) was prepared in DMSO solvent. The solution was then diluted to $1.25 \times 10^{-4}$ M to obtain a suitable absorbance appropriate for the UV-visible titration studies. Anions solutions (A) were prepared in DMSO solvent to achieve the concentration of $1.25 \times 10^{-2}$ M. For the UV-Vis screening experiment, samples were prepared by filling 2.0 mL of **L** ($1.25 \times 10^{-4}$ M) in the quartz optical cell (1 cm path length) and mixing the receptor solution with an anionic solution separately to maintain the A/**L** = 10. In the titration experiment, the cuvette was filled with a 2.0 mL solution of **L**, and then the respective anionic solution was gradually added up to 50 equivalents using a micropipette. Each titration was performed by 15 incremental addition of the anion ($[A]_0/[L]_0 = 0−50$ equiv) to the receptor solution. The changes in the relative UV−Vis absorbance ($I/I_0$) with an increasing amount of anionic solution were analyzed using a nonlinear regression method at room temperature [45].

## 2.4. NMR Studies

$^1$H NMR experiments were performed to evaluate the binding affinity of **L** for anions using their respective tetrabutyl ammonium salts $[n\text{-}Bu_4N]^+A^-$ (where A = $F^-$, $Cl^-$, $Br^-$, $I^-$, $HSO_4^-$, $H_2PO_4^-$, $HCO_3^-$, $AcO^-$, $NO_3^-$) in DMSO-$d_6$. For the NMR titration studies, the initial concentrations of **L** and an anion were used as $[L]_0 = (2 \times 10^{-3}$ M) and $[A]_0 = (20 \times 10^{-3}$ M), respectively. For the screening experiments, samples were prepared by filling 0.5 mL of **L** ($2 \times 10^{-3}$ M) in the NMR tube and then mixing it with an equivalent amount of each anion separately. Each titration experiment was completed by 13 incremental addition of the anion solution (totaling 0.5 mL, $20 \times 10^{-3}$ M) to the receptor solution. The binding constants of **L** for these anions were determined using a nonlinear regression analysis from the progressive chemical shifts in NH signals, where applicable [45].

## 3. Results

### 3.1. Synthesis

The new urea-based anion receptor (**L**) with four NH donor centers was designed and synthesized in pure powder form by a single-step reaction between *p*-xylylenediamine and 3-nitrophenyl isocyanate as depicted in Scheme 2. To be an effective anion receptor, the molecule should have an ideal combination of binding sites and an appropriate cavity size to complement a certain anion (Scheme 1). In general, neutral anion receptors have such functionalities and properties that can bind an anion selectively over other anions, principally by H-bonding and utilizing different other types of non-covalent interactions, such as π-π* interactions, ion–dipole interactions and electrostatic interactions [41]. In

this case, the ligand **L** has four NH groups as active binding sites possessing excellent H-bonding capabilities, suitable to coordinate an anion within the receptor's framework. In addition to the four –NH groups as binding sites, the anionic host **L** has two *m*-nitro phenyl groups, making it an effective candidate for anion recognition and sensing in solution.

*3.2. UV-Vis Spectroscopic Studies*

3.2.1. UV-Vis Screening

UV-Vis screening experiments of **L** with anions ($F^-$, $Cl^-$, $Br^-$, $I^-$, $HSO_4^-$, $H_2PO_4^-$, $HCO_3^-$, $AcO^-$ and $NO_3^-$) have been performed in DMSO using appropriate concentrations of **L** and anions. As shown in Figure 1, a significant change in the absorbance (A) and/or $\lambda_{max}$ has been observed for **L** after the addition of $F^-$, $H_2PO_4^-$, $HCO_3^-$, $AcO^-$, suggesting that the receptor effectively binds an anion possibly through hydrogen bonding interactions. The $\lambda_{max}$ of the free receptor **L** (354 nm) has been shifted to 360 nm ($F^-$), 365 nm ($H_2PO_4^-$), 360 nm ($HCO_3^-$) and 356 nm ($AcO^-$). The bathochromic shifts ($\Delta\lambda$) due to the addition of anions are listed in Table 1.

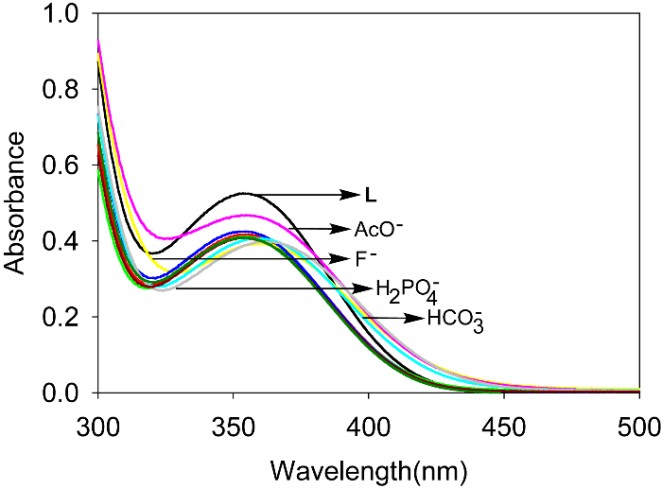

**Figure 1.** UV-Vis screening of **L** ($1.25 \times 10^{-4}$ M) with anions in DMSO.

**Table 1.** Spectral parameters ($\lambda_{max}$) of **L** upon the addition of different anions, as observed from UV-Vis screening in DMSO.

| Sample | $\lambda_{max}$ (/nm) | $\Delta\lambda$ (/nm) |
|---|---|---|
| Free **L** | 354.4 | |
| **L** + $F^-$ | 360 | 5.6 |
| **L** + $Cl^-$ | 355.5 | 1.1 |
| **L** + $Br^-$ | 356 | 1.6 |
| **L** + $I^-$ | 354.5 | 0.1 |
| **L** + $HSO_4^-$ | 355.5 | 1.1 |
| **L** + $H_2PO_4^-$ | 365 | 10.6 |
| **L** + $HCO^-$ | 360 | 5.6 |
| **L** + $AcO^-$ | 355.6 | 1.2 |
| **L** + $NO_3^-$ | 355.5 | 1.1 |

3.2.2. UV-Vis Titrations

To evaluate the binding interactions quantitatively, UV-Vis titration experiments were performed for **L** with the corresponding set of anions in DMSO. Previous work suggests that the nitrophenyl groups functionalized to urea or thiourea receptors are effective chromophores for anion sensing in solution, displaying absorption and colorimetric changes [22–25]. The receptor **L** containing nitrophenyl groups as sensing sites shows an absorption band at $\lambda_{max}$ = 354 nm. The incremental addition of $F^-$ to **L** resulted in a

gradual red shift in the absorbance band (354 to 360 nm) with a gradual decrease in the absorption intensity, displaying an isosbestic point at around 400 nm (Figure 2a). However, for other halides, the absorption intensity decreased without showing an isosbestic point (Supplementary Materials). The change in the absorption intensity of **L** due to the gradual addition of a halide provided the best fit for a 1:2 binding mode [45], showing both 1:1 and 1:2 complexes. The binding constants for 1:1 ($K_1$) and 1:2 ($\beta = K_1 K_2$) are listed in Table 2. Because of the *p*-xylene linker used in **L**, the two urea groups are located on the opposite sides, making the receptor capable of binding two anions.

As shown in Figure 2, similar spectral changes are observed titrating the receptor with acetate, bicarbonate and dihydrogen phosphate, displaying isosbestic points at 382, 398 and 390, respectively. For the addition of nitrate and hydrogen sulfate, the absorption decreases without showing an isosbestic point. The variation of the absorbance as a function of the anion concentration resulted in the best fit for a 1:2 binding model [45] for each of the oxoanions. Selected titration curves and binding constant plots are shown in Figure 2. Additional titration curves and plots are provided in Supplementary Materials. The overall binding magnitude of the receptor does not show much difference for acetate (log $\beta$ = 4.97), bicarbonate (log $\beta$ = 4.58) and dihydrogen phosphate (log $\beta$ = 4.71), which could be due to the flexible nature of the receptor. The binding constants are listed in Table 2, exhibiting the highest binding for acetate among oxoanions with the binding order of acetate > dihydrogen phosphate > bicarbonate > hydrogen sulfate > nitrate, agreeing with the reported binding trend for other neutral receptors [21,22].

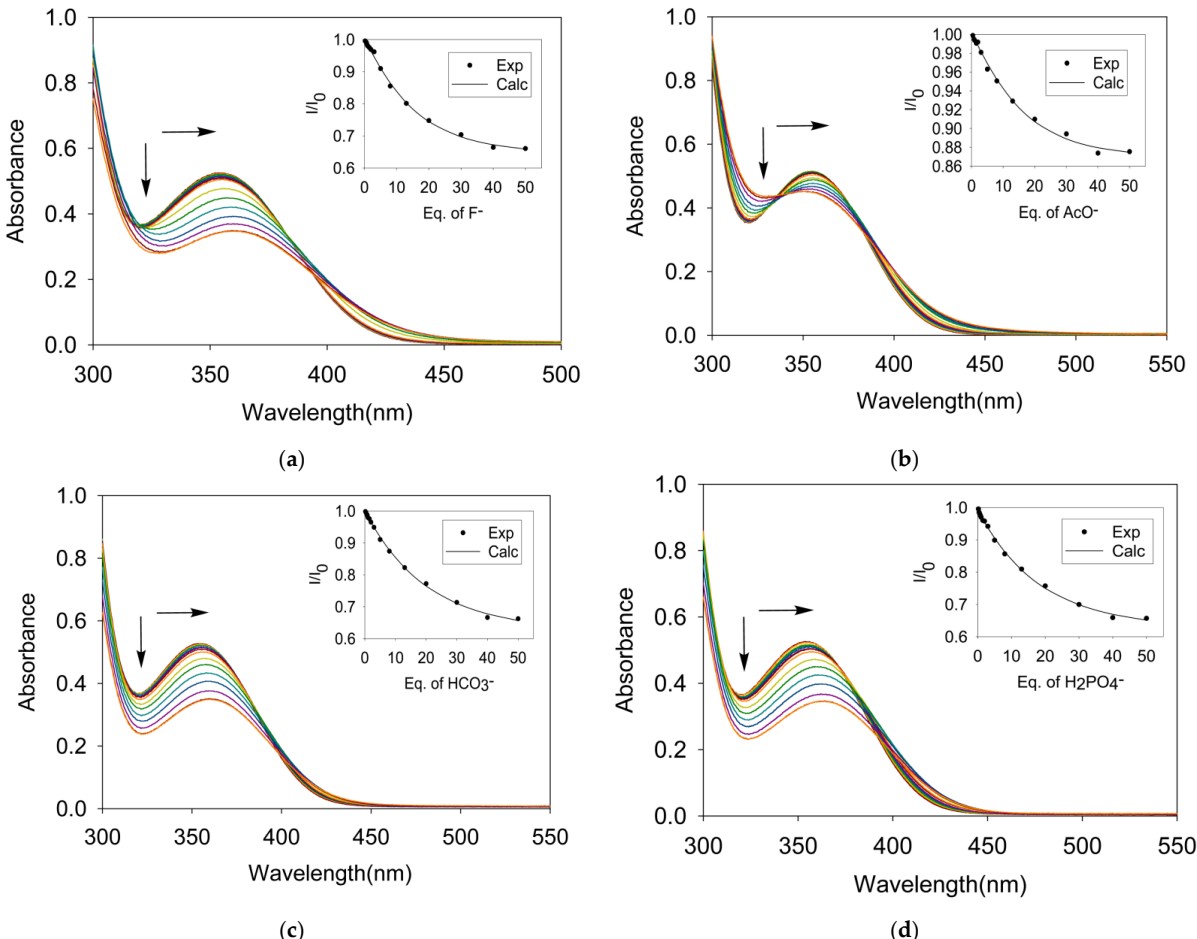

**Figure 2.** UV-Vis titration spectra of **L** ($1.25 \times 10^{-4}$ M) with an incremental addition of (**a**) *n*-TBAF, (**b**) *n*-TBAOAc (**c**) TEAHCO$_3$ and (**d**) *n*-TBAH$_2$PO$_4$ in DMSO. Binding isotherms were calculated using 1:2 binding model.

**Table 2.** Binding constants of **L** for anions.

| Anions | log $K$ (UV-Vis) [a] | | | log $K$ (¹HNMR) [b] | | |
|---|---|---|---|---|---|---|
| | log $K_1$ | log $K_2$ | log $\beta$ = log $K_1 K_2$ | log $K_1$ | log $K_2$ | log $\beta$ = log $K_1 K_2$ |
| F⁻ | 2.12 | 3.12 | 5.24 | [c] | [c] | [c] |
| Cl⁻ | 2.17 | 2.25 | 4.42 | 2.15 | 2.33 | 4.48 |
| Br⁻ | 2.16 | 2.05 | 4.21 | [d] | [d] | [d] |
| I⁻ | 2.27 | 1.87 | 4.04 | [d] | [d] | [d] |
| HSO$_4$⁻ | 2.16 | 1.61 | 3.77 | [d] | [d] | [d] |
| H$_2$PO$_4$⁻ | 2.23 | 2.48 | 4.71 | 2.17 | 2.46 | 4.63 |
| HCO$_3$⁻ | 2.12 | 2.46 | 4.58 | 2.11 | 2.43 | 4.54 |
| AcO⁻ | 1.98 | 2.99 | 4.97 | 2.01 | 2.95 | 4.96 |
| NO$_3$⁻ | 1.99 | 1.55 | 3.54 | [d] | [d] | [d] |

[a] Calculated from UV-Vis titration studies in DMSO; [b] Calculated from ¹H NMR titration studies in DMSO-$d_6$; [c] disappearance of NH signals; [d] negligible change in NMR chemical shifts.

### 3.3. Colorimetric Study

In order to evaluate the applicability of **L** as a colorimetric probe for anions, colorimetric studies have been performed by mixing the receptor with each anion separately in DMSO. As shown in Figure 3, the addition of fluoride, acetate, bicarbonate and dihydrogen phosphate shows a visual color change from off-white to yellow, indicating host–guest interactions. However, no significant color change is observed due to the addition of chloride, bromide, iodide, hydrogen sulfate and nitrate. This observation is consistent with the results of UV-Vis titrations, showing significant bathochromic shifts for fluoride, bicarbonate and dihydrogen phosphate (Figure 1 and Table 1).

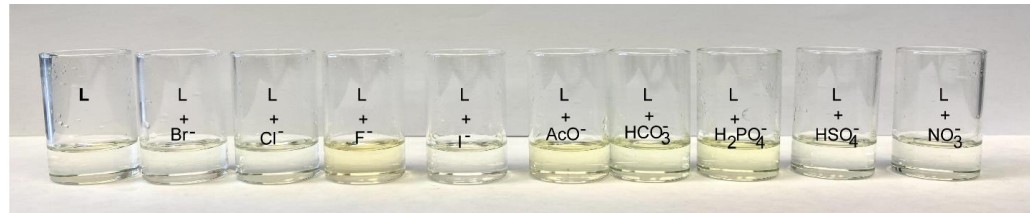

**Figure 3.** Colorimetric studies of **L** ($1 \times 10^{-3}$ M) in the presence of 4 equiv. of anions in DMSO, showing a color change for fluoride, acetate, bicarbonate and dihydrogen phosphate anions.

### 3.4. NMR Studies

Binding interactions between the receptor **L** and anions have also been investigated through ¹H NMR experiments in DMSO-$d_6$ at room temperature using the corresponding tetrabutyl ammonium salts. Figure 4 shows the NMR screening spectra of **L** (2 mM) against various anions, as performed from the addition of one equivalent of an anion to the receptor. Two distinct NH resonances for the free **L** appear at 9.11 and 6.80 ppm in the ¹H NMR spectrum. These NH groups can act as H-bond donors to bind an anion forming a receptor–anion association complex. In addition, the presence of the appended nitro groups in the aromatic rings could enhance the hydrogen-bonding capabilities by increasing the acidity of the NH binding sites. In the ¹H NMR studies, these protons (NH) have closely been monitored by titration experiments. During the screening experiment, it is observed that the NH protons are disappeared after the addition of fluoride, while for chloride, dihydrogen phosphate, bicarbonate and acetate, the protons' resonances have been shifted downfield (higher δ ppm value). This observation suggests that the receptor effectively responses to fluoride, acetate, bicarbonate, dihydrogen phosphate and chloride anions, possibly due to the H-bonding interactions between the receptor and an added anion. However, for other anions (bromide, iodide, hydrogen sulfate and nitrate), the protons' peaks are not observed

to shift significantly (Supporting Materials), indicating weak host–guest interactions under the experimental conditions used for [1]HNMR titrations.

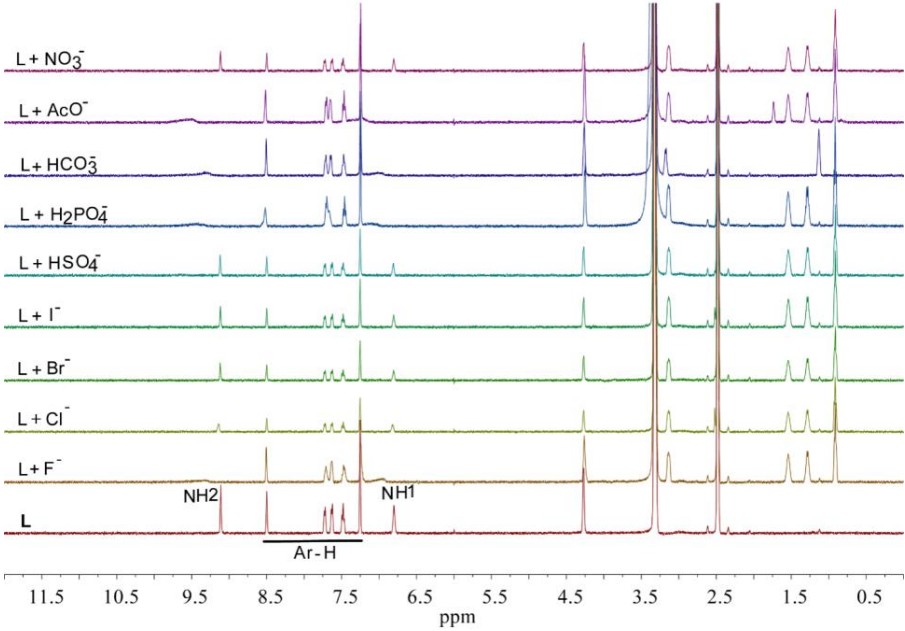

**Figure 4.** [1]H NMR screening spectra of **L** (2 mM) against various anions (20 mM) in DMSO-$d_6$.

In order to estimate the binding constants of **L**, proton NMR titrations have been performed by adding systematic addition of various anions to the receptor using DMSO-$d_6$ as a solvent. Figure 5 shows the stacking of [1]H NMR titration spectra of the receptor upon the incremental addition of fluoride, showing a progressive downfield shift of both NH protons during the first few additions of the anion, demonstrating a typical hydrogen-bonding interaction of the receptor with fluoride. As can be seen, both the –NH resonances have been initially broadened and almost disappeared from the addition of less than one equivalent of fluoride. On the other hand, the aromatic protons showed slight upfield shifts. This observation is consistent with several previous studies, which could be due to the net proton transfer of NH to highly basic fluoride [46–49]. It is possible that the receptor initially binds a fluoride during the first few additions, which ultimately undergoes deprotonation at the higher anion concentrations to form a HF$_2$[-] species [49]. A control experiment with a more basic hydroxide anion has been performed in the presence of fluoride as well as other selected anions. As shown in Figure 6, the addition of one equivalent of hydroxide anion resulted in a complete disappearance of both acidic NH protons and an upfield shift of the aromatic proton resonances due to the deprotonation of NH sites caused by the acid-base interactions [42,49]. This NMR spectrum is almost similar to that obtained from the addition of one equivalent of fluoride to **L**, suggesting the deprotonation of NHs by the added fluoride anion. The addition of hydroxide anion to the mixture of **L** with fluoride also showed the disappearance of NH resonances along with a further upfield shift of aromatic protons. On the other hand, the NH signals that are clearly shown after the addition of one equivalent of acetate, dihydrogen phosphate or bicarbonate, disappear due to the addition of one equivalent of hydroxide anion, suggesting that the binding event is changed to the deprotonation process caused by the highly completive basic hydroxide anion. It has also been reported that a receptor in the presence of a high concentration (e.g., used in [1]H-NMR titrations), the acidic –NH group could undergo a deprotonation process in the presence of a strong basic anion; while under more diluted conditions, an effective anion binding could be observed (as during UV-Vis titrations). Thus, the interactions during the UV titration of the receptor with fluoride could be classified as a binding process.

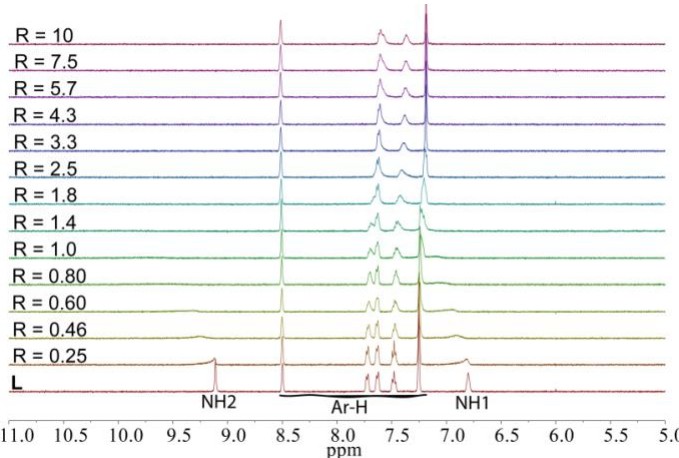

**Figure 5.** $^1$H NMR titration of **L** (2 mM) with the incremental addition of *n*-TBAF in DMSO-$d_6$.

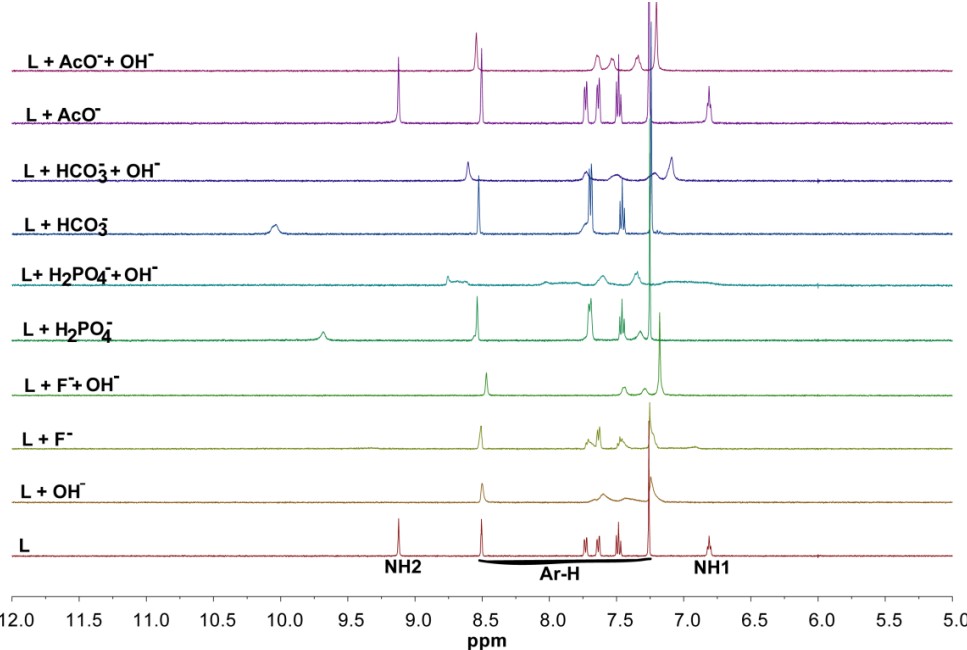

**Figure 6.** $^1$H NMR control experiments of **L** (2 mM) with selected anions (F$^-$, H$_2$PO$_4^-$, HCO$_3^-$ and AcO$^-$) in the presence and absence of *n*-TBAOH in DMSO-$d_6$.

For all other anions, the NH peaks remain visible with the progressive downfield shift of NH protons until the end of the titrations, while the aromatic protons show negligible changes. NH protons are used to calculate the binding constants. The partial $^1$H NMR titration spectra of **L** showing changes in the NH chemical shifts with an incremental addition of acetate, bicarbonate, dihydrogen phosphate and chloride follow similar trends (Figure 7 and Supporting Materials). The variation of NH resonances, as analyzed by the non-linear regression method [45], demonstrates that the receptor binds each of these four anions in a 1:2 binding fashion. On the other hand, an insignificant change in the chemical shift resonances has been observed due to the addition of bromide, iodide, hydrogen sulfate and nitrate, hampering the determination of binding constants. The calculated binding constants as obtained from the NMR titrations studies are listed in Table 2, showing the binding constants (in log $\beta$) of 4.48, 4.63, 4.54 and 4.96 for chloride, dihydrogen phosphate, bicarbonate and acetate, respectively, which are approximately close to those obtained from the UV-titrations. A similar anion binding order was observed by previously reported 4-nitrophenyl-based molecular clefts with an *m*-xylene linker [8]; however, the receptor

L exhibits overall strong affinities for anions because of its ability to bind two anions simultaneously as opposed to one anion that was shown by the reported receptors. A slight variation of the results obtained from the two different techniques could be due to the different conditions used for UV-Vis and [1]HNMR titrations [46]. Nevertheless, the binding order as acetate > dihydrogen phosphate > bicarbonate > chloride that acquired from [1]HNMR titrations parallels with that obtained from UV-Vis, showing the highest binding of the receptor for acetate.

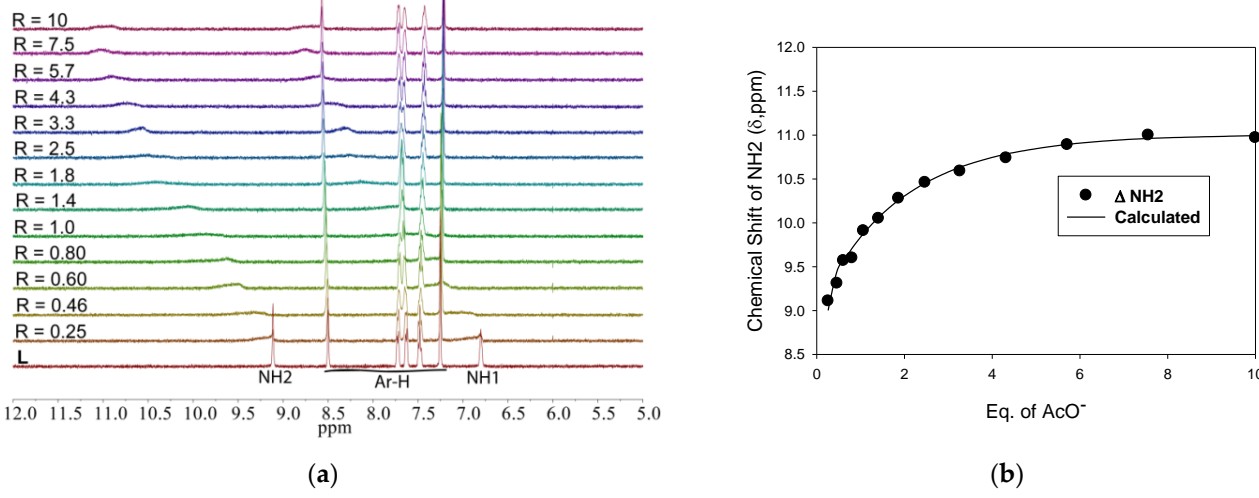

**Figure 7.** (**a**) [1]H NMR titration of **L** (2 mM) with the incremental addition of *n*-TBAOAc in DMSO-$d_6$. (**b**) Titrations plot of **L** with *n*-TBAOAc in DMSO-$d_6$ using 1:2 binding model.

## 4. Conclusions

In conclusion, we have synthesized a simple urea-based dipodal neutral receptor functionalized with a *para*-xylyl-based molecular framework as a spacer along with two nitro groups as signaling units and studied its binding affinity for a wide variety of anions in solution. The results from the titration studies show that the receptor binds anions in a 1:2 fashion, showing the net binding affinity for halides in the order of fluoride > chloride > bromide > iodide and for oxoanions in the order of acetate > dihydrogen phosphate > bicarbonate > hydrogen sulfate > nitrate in DMSO, providing the highest binding for acetate over other anions studied. The observed binding order indicates that the binding is primarily dominated by the relative basicity of anions, reflecting the Hofmeister effect [50]. Additionally, it has been shown that the receptor is capable of displaying a visible color change in the presence of fluoride, acetate, bicarbonate and dihydrogen phosphate in solution. Although the receptor that is based on the simple design and straightforward synthetic strategy efficiently binds a number of common anions in solution, it does not exhibit strong selectivity for a particular anion, which could be due to the lack of a defined cavity. However, this simple compound could be beneficial in an application-oriented area for the detection or liquid–liquid extraction of an anion [51,52]. Furthermore, the results from this study might be useful in designing new molecular chemosensors for selective binding and separation of anions for practical applications.

**Supplementary Materials:** The followings are available online at https://www.mdpi.com/article/10.3390/chemosensors9100287/s1, Figure S1: 1H NMR characterization of **L** in DMSO-$d_6$. Figure S2: 13C NMR characterization of **L** in DMSO-$d_6$. Figure S3: UV-Vis titration of **L** ($1.25 \times 10^{-4}$ M) with incremental addition of *n*-TBACl in DMSO. Binding isotherm is calculated using 1:2 binding model. Figure S4: UV-Vis titration of **L** ($1.25 \times 10^{-4}$ M) with incremental addition of *n*-TBABr in DMSO. Binding isotherm is calculated using 1:2 binding model. Figure S5: UV-Vis titration of **L** ($1.25 \times 10^{-4}$ M) with incremental addition of *n*-TBAI in DMSO. Binding isotherm is cal-culated using 1:2 binding model. Figure S6: UV-Vis titration of **L** ($1.25 \times 10^{-4}$ M) with incremental addition of

*n*-TBAHSO4 in DMSO. Binding isotherm is calculated using 1:2 binding model. Figure S7: UV-Vis titration of **L** ($1.25 \times 10^{-4}$ M) with incremental addition of *n*-TBANO3 in DMSO. Binding isotherm is calculated using 1:2 binding model. Figure S8: 1H NMR titration of **L** (2 mM) with incremental addition of *n*-TBAH2PO4 in DMSO-$d_6$. Binding isotherm is calculated using 1:2 binding model. Figure S9: 1H NMR titration of **L** (2 mM) with incremental addition of TEAHCO3 in DMSO-$d_6$. Binding isotherm is cal-culated using 1:2 binding model. Figure S10: 1H NMR titration of L (2 mM) with incremental addition of *n*-TBACl in DMSO-$d_6$. Binding isotherm is cal-culated using 1:2 binding model. Figure S11: 1H NMR titration of **L** (2 mM) with incremental addition of *n*-TBABr in DMSO-$d_6$. Figure S12: 1H NMR titration of **L** (2 mM) with incremental addition of *n*-TBAI in DMSO-$d_6$. Figure S13: 1H NMR titration of **L** (2 mM) with incremental addition of *n*-TBAHSO4 in DMSO-$d_6$. Figure S14: 1H NMR titration of **L** (2 mM) with incremental addition of *n*-TBANO3 in DMSO-$d_6$.

**Author Contributions:** M.A.H. conceived the project idea. S.K., T.K.E. and Z.Y. synthesized and studied the compound. S.K. prepared the initial manuscript. All authors have read and agreed to the published version of the manuscript.

**Funding:** The project described was supported by the US Department of Defense (Grant Number W911NF-19-1-0006).

**Institutional Review Board Statement:** Not applicable.

**Informed Consent Statement:** Not applicable.

**Data Availability Statement:** All datasets generated for this study are included in the article/ Supplementary Material.

**Acknowledgments:** Analytical core facility at Jackson State University was supported by the National Institutes of Health (G12RR013459).

**Conflicts of Interest:** The authors declare no conflict of interest.

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
