# Peer review of "Spectroscopic and Colorimetric Studies for Anions with a New Urea-Based Molecular Cleft"

_chemosensors, doi:10.3390/chemosensors9100287_

Round 1

Reviewer 1 Report

A urea based dipodal chemosensor for anions was synthesized and studied NMR and UV-visible absorbance spectroscopies in DMSO-d6 and DMSO. Binding constants are measured using both techniques with halide and oxyanions. 

It is noticed that in Figure 4 the NMR proton resonances for acetate, bicarbonate, dihydrogen phosphate and fluoride for the NH1 and NH2 resonances disappear or become broad. A recommendation is for the authors to perform measurements with hydroxide (control experiment) as many anions, notable fluoride, are strong bases in DMSO, which the authors do acknowledge (lines 250-267).   An experiment with hydroxide may give some additional insight into whether the observed binding can be better classified as hydrogen bonding or rather to deprotonation (or perhaps there is a competition between the two events).

The authors are referred to NMR studies in

  1. Mol. Recognit. 2007, 20, 139-144. https://doi.org/10.1002/jmr.816

Eur. J. Org. Chem. 2007, 3999-4010. https://doi.org/10.1002/ejoc.200700294

Org. Biomol. Chem., 2015, 13, 1662-1672. DOI: 10.1039/c4ob02091j

which include measurements of nitro-substituted anion chemosensors with a series of anions including hydroxide.  Acetate was observed to act as a base in these studies.

Abstract: Replace “to detect” with “of detecting”.

Abstract: Add “by” before displaying.

Line 37: Replace “to maintain” with “for maintaining”.

Lines 89-90: Two significant figures are needed for the mmol values.

Table 1: Write “(/nm)”.

Line 292: Replace “verity” with “variety”.

Line 301” Replace “to display” with “of displaying”.

Author Response

We are very thankful to this review for suggesting some critical issues regarding the binding event for fluoride. As recommended, we have performed control 1HNMR experiments using hydroxide , and proposed that the interactions between the host and fluoride lead the deprotonation process while it is a binding event for each of the remaining anions studied. We have elaborated our discussion, provided a new Figure showing the effect of hydroxide anion and cited the suggested references.  We have also fixed some other errors, as suggested.

Reviewer 2 Report

The manuscript by Hossain and co-workers describes the synthesis and evaluation of anion-recognition properties of a new dipodal p-xylyl-based molecular receptor equipped with two 3-nitrobenzene moieties.

The manuscript is well-written and the receptor synthesis is remarkably straightforward. The receptor shows typical selectivity to anions in dmso (acetate > dihydrogenphosphate > bicarbonate > fluoride > halides).

The results are of potential interest to the supramolecular chemistry community. However, the method of determination as well as values of association constants do not convince me. Specifically, the Authors claim that the receptor binds all anions in a 1:1 fashion without providing any evidence to support this claim. This is a rather unexpected assumption, taking into account the molecular structure of the receptor, which is certainly capable of binding two anions at the same time. For example, when the urea groups will be located on the opposite sides of the p-xylene linker. This also suggests that the receptor may undergo self-association, the authors have not written anything about this.

In addition, the Authors used an outdated method for determining association constants (Ref 45).

Currently, there are free tools that allow to validate different binding models (e.g. 1:2 host:guest) and to determine the accurate binding constants from NMR, UV-vis and fluorescence titration experiments (see the excellent paper of Thordarson about this issue: Chem. Soc. Rev. 2011, 40, 1305-1323; these free tools are available on www.supramolecular.org.). In my opinion, the titration data should be recalculated since the provided association constants are unreliable.

Therefore, I recommend publication in the journal Chemosensors after the major review will be provided.

Author Response

We are grateful to this reviewer for excellent suggestions to explore the binding modes of the receptor. We fully agree that the two urea groups on the receptor with a p-xylene linker are located at the opposite sides, making it potential to bind two anions simultaneously.  As advised, we reevaluated our titration data  using the EQNM (Hynes, M. J. EQNMR: a computer program for the calculation of stability constants from nuclear magnetic resonance chemical shift data. Dalton Trans. 1993, 311–312), and found the data fit well with a 1:2 binding model, forming both 1:1 and 1:2 complexes. We have revised our discussion based on the results, added new data in Table 2, and replotted all binding isotherms.

Reviewer 3 Report

The submitted manuscript is well structured and English is acceptable. They propose a simple urea based dipodal neutral receptor functionalized with a para-xylyl based molecular and a priori it is interesting.
The graphics are sufficient and of good quality.
However, the authors must add an essential section to make publication in this journal possible. This section should be related to the potential and possible applications in the field and others about the proposal presented by the authors. In these possible applications they must give details of the advantages or disadvantages that there may be.

Author Response

As suggested by this reviewer, we have added a statement about the possible application of this receptor in the section of conclusions.

Reviewer 4 Report

This work describes the synthesis of the ligand containing a recepting fragment based on two urea-units and two m-nitrophenyl substituents as a chromophore. The authors carried out a thorough study of the coordination properties of this ligand to various anions in DMSO using UV-vis spectroscopy and NMR. The authors demonstrated that the most stable complexes are formed with fluoride and acetate anions. At the same time, the relative stability of the complexes is in agreement with the previously known trends. This work presents the results that could be useful in the development of sensors for anions. Therefore, the work can be published, but significant changes should be made to the discussion of the results obtained.
Notes:
The results obtained seem to be less successful than those previously reported by the authors in ref. [8], in which a similar para-nitrophenyl-substituted ligand exhibited a significant bathochromic shift upon F- binding.
In the discussion, comparison of the results obtained in this work and work [8] should be given. The authors should give an explanation why a very small change in the structure of the ligand led to a decrease in its sensing ability. Perhaps this is due to the conjugation of the nitro group with the NH group of urea in Ref. 8? The authors also should give an estimate of the detection limits for anions using the ligands from that work and ref. [8].
The sentence in conclusion "We have chosen m-nitro substituted benzene ring as terminals, since the attached nitro group acts as an effective chromophore facilitating the desired spectrochemical change with an anion in solution." seems incorrect - the chromophore proposed by the authors showed low efficiency, the spectrum changes are very small. This, apparently, leads to lower sensitivity. The changes are very poorly visible to the naked eye.

Author Response

We appreciate the reviewer for the nice comments and give an opportunity to revise  the manuscript. We have carefully followed the suggestions and made significant changes based on the comments by this reviewer as well as by other reviewers. As we have reevaluated the binding mode with a 1:2 model, the present receptor with a para-xylene linker is capable of accommodating two anions at the two urea groups located at the opposite sides. However, the receptors reported in ref (8) those are both urea and thiourea consisted with meta-xylene linkers which are suitable to bind a single anion within their rigid cavities. We believe, the major difference in binding modes between the present receptor and the reported receptor s is due to the different linkers. We agree with this reviewer that the position of the nitro group(s) may affect the conjugation of the urea groups that may lead to the lower sensitivity. We have revised our conclusions section removing the statement " We have chosen m-nitro substituted benzene ring as terminals, since the attached nitro group acts as an effective chromophore facilitating the desired spectrochemical change with an anion in solution."

Round 2

Reviewer 2 Report

The authors have substantially improved the manuscript, which is now suitable for publication in Chemosensors.
Of the minor comment I noticed - the references 47 and 49 are the same.

Reviewer 4 Report

The authors have significantly improved the manuscript. The article deserves publication.